# Research on the Application of BRBs in Seismic Resistance of Bridge

**DOI:** 10.3390/ma16072549

**Published:** 2023-03-23

**Authors:** Xiaoli Li, Jina Zou, Yuemin Zhao, Dongsheng Wang

**Affiliations:** 1Institute of Road and Bridge Engineering, Dalian Maritime University, Dalian 116026, China; 2Liaoning Key Laboratory of Marine Environment Bridge and Tunnel Engineering, Dalian 116026, China; 3School of Civil and Transportation Engineering, Hebei University of Technology, Tianjin 300401, China; dswang@hebut.edu.cn

**Keywords:** buckling-restrained brace, bridge engineering, energy dissipation capacity, seismic performance

## Abstract

The beneficial effects of buckling-restrained braces (BRBs) in bridge engineering have attracted widespread attention in recent years. Firstly, this paper introduces the basic working mechanism of traditional BRBs, and the new forms and new materials of BRBs are also being studied. Secondly, the responses and performances of BRBs applied to (piers) girder bridges, cable-stayed bridges, and arch bridges are systematically studied. Besides, studies on the connection nodes between BRBs and structures have been paid more and more attention. By comparing and analyzing the damping effect of BRBs alone and that of BRBs with other seismic isolation devices on a bridge, it is determined that a reasonable BRB layout can effectively improve the seismic performance of the bridge with better energy dissipation capacity and load-carrying capacity than other components, but they are less used in practice and do not have mature specifications to be applied on different bridges. Finally, the following trends in BRB development in bridge research are discussed: the diversity of BRB forms, applications of BRB, node connection security, and combined damping measures. These areas should be explored through in-depth theoretical and experimental research.

## 1. Introduction

Modern cities are becoming more and more dependent on transportation with the rapid growth of the population and the development of the economy. As a traffic lifeline, once a bridge is damaged in an earthquake, traffic will be hindered, and restoring and rebuilding the bridge will take a long time, which will affect the daily life of drivers and locals. Therefore, bridge seismic problems cannot be ignored. One of the most widely used damping methods used to mitigate the seismic responses of bridges in recent years is the application of buckling-restrained braces (BRBs). Due to their stable mechanical properties, simple construction, and simple designs, BRBs are effective seismic dampers that are gradually being applied in the study of seismic bridges to improve their seismic capacities. With the development of engineering technology, the structure of BRBs is constantly being updated, and experimental and theoretical research on them has gradually attracted increasing attention.

This paper focuses on the seismic applications of BRBs in bridge engineering, summarizes and analyzes the forms and working principles of new BRBs, and investigates the damping effects of various types of BRBs on different bridges. The use of BRBs as seismic isolation elements in large-span bridges is discussed, and the effects of BRBs, other seismic isolation devices, and their joint application on the seismic performances of bridges are compared. The trends in BRB use and development in the seismic design of bridges are also discussed.

## 2. New Types and Materials of Buckling-Restrained Braces (BRBs)

### 2.1. Basic Composition and Principle

The BRB was originally developed by Yoshino et al. [1] in Japan, who proposed a component form of a steel plate embedded with a shear wall in 1971. Wakabayash [2] first proposed the concept of preventing buckling by constraining the deformation of a plate embedded in the middle of the concrete slab in 1973. The first test was then conducted by Kimura in 1976 [3], which proved that the BRB exhibited good hysteretic behavior. Since then, BRBs have gradually evolved into a form of core that is constrained by peripheral components. Figure 1 shows the steel inner core and outer constrained components (filled with mortar or concrete) of a BRB. During an earthquake, the BRB takes the lead in consuming seismic energy, and the peripheral components protect the inner core from buckling, thus protecting the main components of the bridge and improving the entire bridge’s seismic performance.

### 2.2. Research Status of BRBs

As the damping effect of BRBs becomes more recognized, many scholars have conducted research on the hysteretic performances and damping effects of novel BRBs. Based on the traditional BRB, Chen et al. [4] developed a new ductile assembled buckling-restrained brace (DA-BRB), as shown in Figure 2. The limit baffle on the core plate could effectively control the yield of the core plate step by step and allow the core plate to fully develop plasticity. Through finite element analysis and displacement loading tests, it was found that the performance of the DA-BRB was less affected by the width-to-thickness ratio of the core plate and the number of sections. The axial stiffness, bearing load, and energy dissipation capacity of the DA-BRB increased with the reduction of the wedge ratio (max/min cross-sectional area of the inner core) of the core plate.

Zhang et al. [5] proposed a new type of BRB that had double-restrained square steel pipes (DRSSP-BRB), as shown in Figure 3. The structure of this BRB was simple, and its deformation could be easily observed. Through finite element analysis and displacement loading control tests, it was found that the energy dissipation capacity of this new BRB increased with the decrease in the slenderness ratio, and the hysteresis curve became wider. At the same time, the thickness of the steel tube was 4 mm, and the gap was 2 mm, which could reduce the influence of the frictional force.

BRBs can be divided into single-core and dual-core braces based on the number of cores. Cai et al. [6] first proposed a dual-core BRB that combined two inner cores with two steel tubes. Guo et al. [7,8] studied the design method and hysteretic performance of a dual-core BRB, and based on this, Zeng et al. [9] proposed a prefabricated dual-core BRB with all steel T-type inner cores and outer constrained components (TBRB), as shown in Figure 4. By conducting quasi-static tests on six specimens with different parameters, it was found that the TBRB exhibited good hysteretic performance, and the energy dissipation capacity of the TBRB could be improved by using a bolted connection and a middle-limited part.

The core of a conventional BRB is continuous and homogeneous steel. To make the BRB lighter, Cahís et al. [10] proposed a perforated-core buckling-restrained brace (PCBRB), as shown in Figure 5. Hydraulic tests were conducted by changing the width and geometry of the transverse band to analyze and verify the equations of the in-plane wavelength and out-of-plane wavelength under a high-mode buckling state. It was found that the new BRB was lighter, the inner core was easier to replace, and the rigidity increased with the ratio of the hollow radius to the side plate width. Subsequent tests could be conducted with a larger specimen size to verify its validity. Zhou et al. [11] carried out an experimental study on PBRB and provided the design process. The only difference between PBRB and PCBRB was that the perforated inner core of PBRB was rectangular, which had the same advantages.

Shi et al. [12] studied a toggle BRB system that could amplify the effect of the damper, as shown in Figure 6. The system could amplify the relative axial displacement of the BRB to achieve a higher energy dissipation capacity. The accuracy of the brace parameters was verified by nonlinear analysis of both the bending damage and bending–shear damage modes. The design process ensured that the main components of the reinforced bend would remain within the elastic range under design-level earthquakes, and it was found that key parameters, such as the displacement amplification factor and the ratio of steel core length to the total length, could be easily obtained using this design system.

Hu et al. [13] proposed a new type of “cross-like” double-yield buckling restrained brace (DYBRB), and the inner core of DYBRB was welded by three plates. The results showed that DYBRB could dissipate corresponding energy much more than traditional BRB under different levels of an earthquake.

Wei et al. [14] proposed the novel central buckled structure shown in Figure 7 and compared the seismic performances of a suspension bridge under three configurations (common central buckle, BRB central buckle, and BRB central buckle with viscous damper (VD)). Through dynamic characteristic testing, they showed that for a suspension bridge with a BRB central buckle, the stiffness of the suspension bridge improved, and the longitudinal beam displacement and bridge tower force decreased. However, the truss stress near the central buckle of the BRB increased with the increase in the BRB yield force, and it was found that the combined application of the BRB central buckle and VD was the best choice to reduce the seismic responses of long-span suspension bridges.

BRB central buckles of different yield stresses were analyzed based on the IDA analysis and earthquake vulnerability analysis, which was introduced by Liu et al. [15]. The results showed that BRB central buckle with appropriate yield stress owned good restriction and energy dissipation capacity with the lowest probability of damage to the tower.

Guo et al. [16] proposed a triple-truss-confined BRB (TTC-BRB), as shown in Figure 3. TTC-BRB possessed excellent hysteretic response under cyclic loads with the advantages of long-span and load-carrying capacity. Wang et al. [17] proposed a steel bamboo-shaped energy dissipater (SBED) which consisted of an inner bamboo-shaped core and an outer restraining tube. By conducting hydraulic tests and finite element analyses of SBED, SBED demonstrated stable and repeated hysteretic capacity. However, the lengths of the segments and the stress concentration around the fillet will affect SBED and need focusing. Arash et al. [18] studied the performance of BRB, TTC-BRB, and SBED. Finite element analyses demonstrated that TTC-BRB showed the best performance considering the gap between the core and the casing and the initial imperfection. Different BRBs have different design parameters, which need further study.

### 2.3. Study on Self-Centering Buckling-Restrained Brace (SC-BRB)

A BRB consumes energy through the tension and compression of the steel inner core, and it undergoes large residual deformation by the plastic accumulation of steel. To solve the shortcoming of the large residual deformation, a self-centering system is proposed. A self-centering system is one of the important systems of earthquake-recoverable functional structures [19,20]. Christopulos et al. [21] first used the internal viscous resistance of the component to consume energy and proposed the concept of a self-centering energy dissipation brace (SCEDB) that was composed of composite materials. The system consisted of two bracing members, a tensioning system, an energy dissipation system, and a series of guiding elements. In addition, two bracing members interacted with the tensioning system. A dissipative mechanism was connected to the two bracing members and was activated when bracing members came into play. The SCEDB for steel structures was verified to have little residual deformation and did not usually change after an earthquake [22,23]. Zhou et al. [24] proposed a self-centering braced rocking frame (SBRF) system, which had a large stiffness and large self-centering capacity. This system can carry horizontal force and eliminate residual displacement through a self-centering brace, and control key node damage through a rocking mechanism. A series of research works were carried out on the prestressed rocking pier structure with different forms of energy dissipation devices. The dampers can be replaced directly after damage, which greatly improves the post-earthquake repair ability of the rocking system [25,26,27].

A self-centering-steel buckling-restrained brace (SC-SBRB) is composed of a buckling-restrained system and a combined disc spring self-centering system, which was introduced by Xu et al. [28], as shown in Figure 8. Under tests with a low-cycle reciprocating load, the SC-SBRB had a high energy consumption and self-centering capacity and little residual deformation when the disc spring had sufficient precompression.

Han et al. [29] developed a disc-spring self-centering buckling-restrained brace (DS-SCB) and established three working conditions: the BRB, self-centering brace (SCB), and DS-SCB, as shown in Figure 9. Based on the comparison of the hysteresis curves and residual displacements, the DS-SCB had a strong energy dissipation capacity and little residual deformation under the pseudo-static experiment. The DS-SCB was a stable and effective damping element due to the friction between the disc spring and the inner sleeve.

A new self-centering buckling-restrained brace (SC-BRB), consisting of a self-centering system and a traditional buckling-restrained system, was introduced by Dong et al. [30] and, as shown in Figure 10a. During the quasi-static cycle test, the SC-BRB showed a flag hysteresis response and medium energy consumption capacity with little residual deformation, as shown in Figure 10b. At the same time, an SC-BRB was installed on each pier of a typical double-column pier bridge by nonlinear analysis. Compared with the BRB, the SC-BRB showed better hysteretic behavior and reduced the residual displacement and the bridge peak acceleration.

Chou et al. [31] proposed a self-centering sandwiched buckling-restrained brace (SC-SBRB), as shown in Figure 11. Some cyclic tests showed that the SC-SBRB exhibited a stable hysteretic behavior, a high capacity for energy dissipation, self-centering characteristics, and little deformation. It also proved that the SC-BRB has good development prospects.

### 2.4. Study on New Materials of BRBs

Shape Memory Alloys (SMAs) are novel metals with distinct features and desirable potential to reduce deformation. SMAs have well hysteretic behavior, excellent re-centering capability, and large damping capacity, and they have been designed to use in structural vibration control and seismic isolation devices [32,33]. Qiu et al. [34] evaluated the dynamic responses of the Steel BRBFs and FeSMA BRBFs under earthquake ground motions. Using nonlinear static analysis and nonlinear time history analysis, it showed that FeSMA BRBs could leave smaller residual deformation than steel BRBs. FeSMA has excellent low-cycle fatigue performance, higher fatigue life, lower cost, and larger energy-dissipation capacity. Shan et al. [35] established a buckling-constrained brace model and added SMA materials to the design. SMA BRB was installed between the piers and the cover beam, and based on the time-history analysis of a girder bridge, it was found that SMA BRB could reduce the bending moment of the bottom of piers, the displacement of the top of piers and beams. SMA BRB was more effective in reducing the seismic response under the E1 earthquake.

Zhao et al. [36] put forward a new type of brace, the maintenance-free steel-composite buckling restrained brace (MFSC-BRB). The inner core was steel, and the restraint unit was a ribbed glass fiber-reinforced polymer (GFRP) rectangular tube. Under the action of a low weekly repeated loading test, it was found that MFSC-BRB had good integrity and light weight and good energy dissipation capacity. Moreover, MFSC-BRB was suitable for high-rise buildings and bridge engineering. Li et al. [37] proposed using double A5083 aluminum alloy inner cores of BRB. The BRB had the advantages of being lightweight and low-cost with well energy consumption capacity. Yang et al. [38] proposed a combined angle steel BRB with different filling materials and unbonded materials. The results showed that the lightweight aggregate concrete made BRB lighter, and the friction made the performance of BRB worse.

In summary, the construction of traditional BRBs has been continuously updated and developed. When BRB plays its energy dissipation role, the inner core is the main force-bearing component, so the form and material of the inner core are the focus of research. New BRBs still have the parts of the inner core and the outer constraint unit, and some of the inner filling material is omitted to achieve a lighter effect. Some add springs and other components to give them the ability to reset. The inner cores are lighter and easy to change. As shown in Table 1, new BRBs show an excellent energy dissipation capacity than traditional BRBs. BRBs also have other disadvantages; only the most influential friction is listed in the table. All steel BRBs show excellent performance in load-carrying capacity and energy-dissipating capacity.

The length and thickness of the core plate are the focus of attention in the experiments. The energy dissipation capacity of BRBs increased with the decrease in the width-to-thickness and the slenderness ratio. More tests are needed on the parameters of BRBs. For BRBs with self-centering ability, the increase of precompression and stiffness of disc spring will increase the load-carrying capacity of the BRBs, but attention should be paid to local deformation. Through theoretical and practical research, BRBs have gradually become reliable damping elements and are expected to be widely applied in practical structures. 

In the design of BRB, Technical Specification for Seismic Energy Dissipation of Buildings (JGJ 297-2013) stipulates the value of the common core section size of the BRB: for the one-shaped core, the width-thickness ratio is 10–20; cross-shaped core, the width-thickness ratio is 5–10; the diameter-thickness ratio of pipe section should not exceed 22. The American Seismic Code AISC (341-10) stipulates the performance tests of BRB: the axial displacement is required to carry out two cycles of tension and compression cyclic loading, and then the fatigue loading is carried out with the middle axial displacement until the bearing capacity of the member decreases, and the cumulative plastic strain in the process is required to reach 200 times the yield strain. BRB load-carrying capacity and energy dissipation performance, the yield strength of steel core is closely related to the seismic grade of the structure and the fortification intensity of the area. It is necessary to grasp the overall layout and performance of the structure and constantly improve the overall seismic performance by calculating.

## 3. Study on Seismic Performances of BRBs

### 3.1. Seismic Performances of BRBs on Piers and Columns

A bridge pier is not only the main component used to bear a superstructure but also the main component that bears the structure’s seismic inertial force. Significant damage to the bridge pier will lead to the collapse of the bridge, which will be difficult to repair quickly after an earthquake. If BRBs are installed on bridge piers, the BRBs will preferentially consume energy to mitigate the seismic response, thereby preserving the integrity of the pier [39].

Li et al. [40] proposed a method of setting BRBs on regular and irregular bridge piers of double-column bridges in mountainous areas and studied the damping effect of BRBs through spectral analysis, nonlinear analysis, and incremental dynamic analysis. It was found that the BRBs could reduce the bearing forces of the bridge piers during small earthquakes but increase the foundation shear. During large earthquakes, the damping rates of the bridge piers with BRBs decreased with the increase in the peak seismic acceleration, and the residual displacement also decreased accordingly. The BRB design with a diagonal multilayer arrangement on 60-m double-column high piers that was introduced by Xie et al. [41] could achieve 40% and 50% damping effects on the curvature of the pier bottom section and the displacement of the pier top, respectively. Dong et al. [42] studied the influence of the BRB arrangement and yield strength on the seismic responses of bridges with different pier heights using nonlinear analysis, and the results showed that when the pier height was approximately 9 and 18 m, the single-inclined-pole BRB and the double-parallel-inclined-pole BRB were the optimal arrangements, respectively, and achieved the best seismic performances. The core section of the BRB on the different high piers of bridges should be steel with different yield points. Liu et al. [43] studied the optimal placement principle of BRBs in replaceable high piers. The results showed that BRB arranged in the area above 1/2 pier height could better play the role of damping energy dissipation, and the arrangement of BRB should consider the principle of lateral stiffness ratio of the pier column. Zheng et al. [44] studied the damping effect of BRB location, installation angle, and quality on the tall piers of continuous rigid frame bridges. The results showed that BRB installed in the middle of the piers could improve the longitudinal seismic performance of the bridge.

The damping effect of BRBs arranged on the bridge bents with double piers was studied by Shi et al. [45,46,47] using nonlinear analysis, and it was found that the seismic damage of regular and irregularly framed bent piers could be effectively reduced when the horizontal yield displacement ratio of the BRB to the shelf pier was 0.5–1.5, and the horizontal stiffness ratio was 0.5–2. Sun et al. [48] used incremental dynamic analysis and quasi-static tests to set the BRB on bridge bents and found that the core section of the BRB would affect its yield strength, which could delay the failure process and improve the stiffness of the bridge bents. El-Bahey et al. [49] applied BRBs on the seismic reinforcements of curved bridge piers to improve their stiffness and strength and found that the BRBs kept the piers elastic through hysteretic behavior to dissipate earthquake energy. Zhang et al. [50] selected an elevated highway double-column pier system with a BRB as the research object and studied the working principle of the BRB through a nonlinear analysis method. They found that the axial force of the pier could be reduced by repeated energy dissipation after BRB yield. When two single-diagonal brace BRBs with an equivalent cross-sectional area of 4000 mm^2^ were selected, the bending moment and shear damping rate of the pier bottom reached 40% and 35%, respectively.

Through a lot of research, the optimum mechanical parameters of BRB can effectively protect the column piers and improve the seismic performance of the bridge substructure. When BRB is arranged between piers, it is mainly arranged laterally. BRB has obvious damping effect because BRB is easier to enter the yield state and plays the role of energy dissipation under horizontal earthquakes. When the height of the pier is different, the BRB dissipates energy by changing the transmission path of the force, and the damping effect of the high pier is better than that of the low pier. The arrangement and quantity of BRBs need to be determined according to the form and height of the pier.

### 3.2. Seismic Performances of BRBs on Girder Bridges

The girder bridge is a very important type of modern bridge, which is the most basic bridge type. Many scholars have conducted experimental and theoretical research on BRB arrangements in beam bridges as a seismic measure.

Carden et al. [51,52] proposed using BRB as ductile end cross frames and compared it against the performance of yielding X braces of a steel girder bridge, as shown in Figure 12. Through the cyclic loading test, the BRBs were less likely to need replacing than X braces after an earthquake with better energy dissipation.

Upadhyay et al. [53,54] proposed the use of a curved beam bridge as a background and arranged BRBs and SCEDBs between the cover beam and the pier. The performances were compared through nonlinear analysis and incremental dynamic analysis under far-field and pulsed earthquakes. It was found that both the BRBs and SCEDBs improved the seismic performances of the bridges, and the SCEDBs had significant advantages in reducing the residual displacements of the bridges. Bazaez et al. [55,56] conducted tests and numerical studies on a BRB in a reinforced-concrete anti-bending bridge and proved its effectiveness. Xu et al. [57] studied the seismic response of a typical double-column curved high-pier bridge and analyzed the efficiency of using a BRB by arranging the BRB between tie beams. Using nonlinear and vulnerability analysis, they showed that the BRB could effectively reduce the seismic vulnerability and improve the performance of the original bridge regardless of whether the response was linear or nonlinear. Wang et al. [58] selected a box-girder bridge as an example and arranged a BRB between the cover beam and the pier in Figure 13. Through nonlinear analysis, they found that the BRB could improve the seismic performance of the bridge under normal and extreme-use conditions by reducing the bending moments, displacements, and strains of the concrete columns and reducing the potential damage to the shear keys between the columns and abutments. Besides, BRBs improved the life safety of bridges.

Ijan et al. [59] tested a resilient post-tensioned hybrid bridge bent with a diagonal BRB between the cap beam and the footings. The system, which had BRB of a resilient post-tensioned bridge bent, was effective and could make the bridge restore quickly. Then, Ijian et al. [60] tested a post-tensioned bridge bent with BRB (PT-BRB), which could tolerate a much higher earthquake demand.

Dong [61] proposed the design concept of applying BRBs and SCEBs on a double-column bridge, with a reinforced-concrete double-column bridge as the research object. Through quasi-static tests and nonlinear analysis, it was found that the SCEBs could improve the transverse stiffness of the bridge and the energy dissipation capacity and also protect the main pier structure and bridge from suffering damage and undergoing residual displacement. Celik et al. [62] proposed the concept of a bidirectional ductile end diaphragm system (EDS) and arranged BRBs on the end diaphragm of a straight steel bridge, as shown in Figure 14. The results showed that the two schemes could effectively resist the longitudinal and transverse seismic forces, and that scheme A had a lower yield force and simpler connections than scheme B. Based on EDS, Xiao et al. [63] designed and tested two types of BRBs with pin-end connections of a single-span steel slab-on-girder bridge. The results showed a recommended design procedure of EDS in both skew and nonskew bridges to ensure BRB performance. Xiao et al. [64] then studied the effect of temperature changes on the BRB design in EDS. By low-cycle fatigue analyses, the minimum ratio of the BRB length over the bridge length should at least be 6% to satisfy the bridge 75 years of design life.

### 3.3. Seismic Performances of BRBs on Cable-Stayed Bridges

As one of the main bridge types for large-span bridges, cable-stayed bridges have large spanning capacities and beautiful structures. During an earthquake, most of the inertial forces of the main girder of a large-span cable-stayed bridge are transmitted to the towers through the stay cables, resulting in increased movement at the base of the tower, accompanied by excessive main girder displacement, which can easily cause bridge instability. The introduction of BRBs into cable-stayed bridges as a damping measure has been proposed as a new seismic system.

Sun et al. [65] proposed a method of using energy-dissipating auxiliary piers to control seismic damage of long-span cable-stayed bridges. It was found that the higher the yield strength was, the smaller the dip angle and the larger the cross-sectional area of the BRB were, and the stronger the energy dissipation capacity of the auxiliary pier would become. With the background of a concrete cable-stayed bridge with two towers and two cables under construction, and a low center of gravity, Zhang et al. [66] proposed eight combined earthquake-resistance measures by changing the pier column section and BRB forms in Figure 15. Nonlinear analysis showed that the BRB improved the transverse seismic performance of the side span pier column, and the scheme in which the side span pier column was changed into a double-limb section and the BRB was arranged across the section had the best seismic performance.

Li proposed a new energy dissipation method of setting BRBs in the longitudinal bridge direction between the tower (pier) and a beam of a sea-cable-stayed bridge [67]. Chen et al. [68,69] compared the seismic responses of an original bridge structure to the same bridge with a vicious damper or a BRB, as shown in Figure 16. It was found that the BRB could control the displacement of the bearing and the top of the tower and greatly reduce the bending moment of the tower (pier) compared with the VD.

### 3.4. Seismic Performance of BRB on Arch Bridges

Ordinary seismic isolation bearings are not suitable for special bridge types, such as long-span arch bridges, so BRBs are novel damping structures for arch bridges. Usami et al. [70] first conducted a cycle test on BRBs and determined that the maximum strain the BRB could withstand was greater than 20 times the yield strain of steel. With the background of a steel arch bridge, BRBs were installed in a herringbone pattern on the columns and main arch in Figure 17. Nonlinear analysis showed that the seismic performance of the steel arch bridge was improved effectively. Based on this bridge, Chen et al. [71] performed dynamic analysis by inputting ground motions one and three times. They found that the BRB installed near the top of the side piers and the arch ribs could withstand repeated earthquakes and that the BRB could fully consume energy and improve the seismic performance of the arch bridge.

Zhang et al. [72] selected a long-span steel truss railway arch bridge spanning a V-shaped canyon, arranged a speed-locking device and a VD at the junction pier in the longitudinal direction, and set K-shaped BRBs at the bottom and top of the chord planes in the main arch ring in the transverse direction. The nonlinear analysis showed that the VD made the bending moment and shear damping rate of each bar in the arch springing section reach 10%, and the BRB resulted in up to a 20% reduction of the internal forces at the bottom section of the columns on each arch. Li et al. [73,74] studied the aseismic problems of arch bridges and proposed a method with BRBs instead of beams or transverse braces to form an energy dissipation system under strong earthquakes in Figure 18. In this paper, three layout methods (near the arch springing, at the top of the arch, and using the arch springing to fix the independent K-brace of the beam) were introduced, and the application prospects and values of the BRB in the energy dissipation and damping design of the arch bridge were emphasized. Gao et al. [75] applied BRBs on a steel truss arch bridge. The elastic-plastic time history analysis showed that BRB could reduce the internal forces and displacements of the arch ribs by replacing a portion of the normal bars. Shao et al. [76] studied that BRB arranged between bent piers could improve the lateral seismic performance of light, flexible arch bridges under long-period ground motions. The damping effect of BRB will be influenced by the types of ground motions and the yield strength of BRB. However, BRB might increase the damage of high piers, and the vertical ground motions should be considered.

Table 2 shows the layouts of BRBs on different bridges, and BRBs had obvious advantages in improving the bridges’ seismic performance. BRBs also had little disadvantages, which could be improved without having a large impact on bridges. BRB is a new way to build seismic bridges, which gives the bridge a recoverable function. Due to the special complexity of the bridge structure, there is no common method of BRB in the bridge, and it is necessary to study separately for different bridges. The seismic capacity of the bridge can be calculated mainly with the methods of the response spectrum, time history analysis, and pushover analysis [77]. Through the analysis results, the weak parts of the bridge are obtained and reinforced by BRB.

### 3.5. Study of BRBs on Actual Bridges 

After decades of development, BRBs have transitioned from laboratory research to practical engineering applications. Some countries and regions in the world have already applied BRBs in projects, which has played an exemplary role in the popularization of BRB applications in bridge seismic structures.

Taking Yong Ning Yellow River Bridge (Figure 19) as an example, Guo et al. [78] studied the seismic response regularity and reasonable damping system of a long-span concrete cable-stayed bridge under a strong earthquake. Through nonlinear analysis, the elastic cable with a VD was used at the connection between the tower and beam, and a BRB was used at the connection between the pier and beam. It was found that the BRB reduced the bending moment of the bridge transverse pier by more than 75%, which proved the necessity and effectiveness of the BRB.

Monito Bridge [80,81,82] is located on the Osaka line of the Hanshin Highway in Japan. It is the third longest truss arch bridge in the world, with a total length of 980 m, as shown in Figure 20. By replacing the diagonal braces on the original cross-linked main tower and the vertical braces of the main tower with BRBs, the maximum reductions in the internal forces of the upper and lower chord bars were determined to be 85% and 42%, respectively, through dynamic analysis.

Wangdu Bridge [80,81,82], Hiroshima Prefecture, Japan, has a span of 99 m. The location of the BRBs on the bridge is shown in Figure 21. It was found that the stress ratio of all the members was reduced by more than 50%, and the BRBs improved the seismic performance of the arch bridge through dynamic time-history analysis.

Luding Dadu River Bridge [83] is a suspension bridge with a total length of 1100 m. The BRB was arranged on the layout of the BRB on the bridge, as shown in Figure 22. The nonlinear analysis showed that the combination of a BRB as the central buckle and a VD in the high seismic area could effectively mitigate the seismic response of the long-span bridge. The longitudinal displacement of the main beam could be reduced by 76% at most, and the shearing force of the main tower could be reduced by 30%.

### 3.6. Research on BRBs at the Nodes

The nodes where the BRB is connected to the bridge will restrain the bridge under an earthquake and should theoretically have some energy dissipation capacity. Reasonable and reliable nodal connections can improve the overall seismic performance of the bridge and should be considered in the theoretical analysis. There is less research related to the connection of the BRB to the bridge, and similar connections in building structures should be studied.

Zhang et al. [84] proposed semi-rigid nodes connected by bolts and angle steels instead of the original rigid nodes of the BRB. Quasi-static tests of three steel specimens with different thicknesses showed that the energy dissipation capacity of beam-to-column nodes increased with the number of semi-rigid joints, but the energy dissipation capacity of the BRB decreased with the influence of the semi-rigid nodes.

Zhao et al. [85] proposed to adopt a new type of sliding anchorage plate node in the soldering between the BRB and the frame, as shown in Figure 23. The beam and column side pre-buried parts were positioned by positioning angles, positioning plates, and bolts, and then they were plug-welded to the anchor plate. The positioning plates and positioning angles were removed after pouring and maintaining the concrete. Finally, the nodal plate was connected to the anchor plate by a welded seam. A quasi-static test showed that the new node effectively improved the seismic performance of the BRB-RC frame and effectively reduced the beam bending deformation in the joint plate area and the influence on the BRB deformation.

Bai et al. [86] proposed a perfobond strip connector (PBL) gusset plate connection and applied it to RC frame structures. The PBL joint plate can be divided into two parts according to its function. The first part is the external part corresponding to the traditional joint plate, which is exposed outside the beam and column concrete and directly connected with the BRB; The second part is the part embedded in the concrete of the beam-column, which mainly transmits the BRB axial force from the external joint plate through PBL, as shown in Figure 24. The results showed that PBL could effectively improve the bearing capacity and seismic performance of the node.

Li et al. [87] proposed an end directly connected triple steel tube buckling-restrained brace (EDTBRB). The results showed that the direct end connection improved the end stress and effectively avoided the yield or failure of the joint section before the core unit. 

The new joint can effectively release the tangential constraint between the joint plate and the sub-frame, reduce the opening and closing effect, the shear force, and plastic damage of the sub-frame beam and column, and give full play to the energy dissipation effect of BRB. Therefore, it reflects that more attention should be paid to the node problem when BRB is connected to the bridge to ensure the stability of BRB and bridge safety. The design of the node needs to be determined according to the specification.

## 4. Damping Effect of BRB

### 4.1. Comparison of BRB with Other Seismic Isolation Components

In recent years, seismic isolation devices have been widely used in bridge engineering [88,89,90,91]. Common seismic isolation devices include bearings (such as lead rubber bearings and sliding friction bearings), dampers (such as fluid VDs and BRBs), and limiting devices. The seismic isolation device is divided into damping and isolation methods, and the isolators will first enter the plasticity to produce a lot of damping and consume a lot of energy entering the structural system; the dampers will prevent seismic energy from entering the structure. The two working principles are different, but they will weaken the impact of the earthquake on the main structure. BRBs, as braced and energy-dissipating dampers, have demonstrated superior seismic performances compared to other seismic isolation devices.

Marco et al. [92] proposed fast design procedures for isolation systems. According to the design period and design equivalent viscous damping, three individual isolation systems (Low and High Damping Rubber Bearings, Lead Rubber Bearings, and Curved Surface Sliders) were designed. The nonlinear time history analyses demonstrated that the design procedure of a building was effective in the isolator peak displacement demand and the building base-shear response. Alper et al. [93] used curved surface sliders (CSS) in an elevated silo group. The incremental dynamic analysis demonstrated that CSS reduced the response of all parameters and the collapse risk under strong earthquakes. Young et al. [94] used rubber friction bearing (RFB) and BRB systems in the same frame. Numerical results through nonlinear time-history analysis showed that the combination of isolators and BRB systems was a good choice to safeguard the structure and minimize damage under earthquakes. Afshin et al. [95] used nonlinear viscous dampers (NVDs), which are arranged on the first two panels from each side of the arch and connected to the truss layers. Nonlinear analysis showed that by using the proposed damping correction factor, the mechanical properties of NVDs could be selected, and the seismic requirements of the bridge could be satisfied. Li et al. [96] proposed a hybrid isolation system consisting of a BRB and a VD. This system effectively dissipated energy and protected a high-rise building under the actions of earthquakes and winds. Moreover, BRBs were combined with isolators to simultaneously mitigate the seismic responses of bridges.

Shi et al. [12] proposed a toggle BRB system that combined the structural fuse concept with toggle brace mechanisms. The system could improve the energy dissipation capacity of the BRB and keep the RC bridge bents elastic. Liu et al. [97] proposed two different hybrid isolation systems (RB (Rubber Bearing)–BRB and LRB (Lead Rubber Bearing)–BRB) on bridge piers. Based on the nonlinear time history, results demonstrated that the LRB–BRB was the most effective isolation system. Guo proposed et al. [98] a new lateral isolation system composed of elastic cables and fluid viscous damper at the girder–tower connections, and BRBs were used for lateral isolation of the piers. Numerical results through nonlinear time history analysis showed that the new system could properly control the seismic responses of the bridges. Li et al. [99] studied three different systems (without braces, with SC-BRBs, and with lock-up self-centering buckling-restrained braces (LU-SC-BRBs)), taking a continuous beam bridge as the background, as shown in Figure 25. Through modal analysis, it was found that the LU-SC-BRB could effectively control the pier bottom bending moment, shear force, and pier top longitudinal displacement. The LU-SC-BRB had a stronger energy dissipation capacity and a longer life cycle than other systems, and it improved the seismic capacity of the bridge.

Joel et al. [100,101] studied the feasibility of installing BRBs on the Vincent Thomas Bridge to reduce the maintenance cost of the bridge by replacing the problematic VDs of the original bridge. The results showed that the BRBs were effective on long-span bridges. Upadhyay et al. [102] conducted in-situ quasi-static tests on a bending bridge with two seismic retrofitting schemes involving BRBs and SCBs that were arranged diagonally on the piers. Nonlinear analysis and incremental dynamic analysis were used to compare and evaluate the performances of the two schemes and the original bridge. The results showed that both schemes improved the seismic performance of the bridge and that the BRB had a better effect in reducing the peak displacement and achieved greater cumulative energy dissipation than the SCB. However, the SCB was better at reducing the residual displacement and had a lower maintenance cost than the BRB while also improving the elasticity.

Montazeri et al. [103] studied three different seismic measures: the lead rubber bearing (LRB), friction pendulum system (FPS), and BRB (installed between three-column piers) against the background of a four-span girder bridge using nonlinear analysis and seismic fragility analysis. Based on the results, they found that all of the seismic measures effectively improved the stiffness of the bridge and that the BRB could effectively mitigate the seismic response under a strong earthquake with a high maintenance cost. Dong et al. [104] installed SC-BRBs and BRBs on reinforced-concrete double-column bridge piers and conducted large-scale quasi-static cyclic loading tests to study their hysteretic behaviors. The study found that the SC-BRBs significantly improved the strength and stiffness of the piers and reduced their residual displacements.

### 4.2. Combining BRBs with Other Seismic Isolation Components

It is crucial to choose effective seismic isolation devices in seismic design. Combining BRBs with other seismic isolation devices can meet the damping needs of bridges at different locations and orientations.

Liu et al. [105] proposed a bidirectional seismic isolation system with BRBs in the longitudinal direction and LRBs in the transverse direction with double-column piers. Their study found, through nonlinear analysis, that the combined application system was superior to the separately arranged system, which could effectively protect the piers by reducing the plastic deformation and residual displacement angle of the piers. Liu et al. [106] proposed the joint damping measures of arranging BRBs between the bottom of the main beam and the bent cap of the curved bridge and arranging lead rubber bearings at the bent cap of the pier. Three operating conditions were compared under near-fault ground motion: without BRBs, with BRBs placed between the side pier girders, and with BRBs placed between the middle pier girders. It was found that the joint damping measures with BRBs installed in the middle pier and lead rubber bearings at the abutment of the curved bridge had the best damping effect, which could effectively reduce the displacement of the main beam and the possibility of falling beams as well as mitigate the seismic response of the pier. Guo et al. studied the damping measures of a VD and a BRB center buckle on a suspension bridge. The truss stress and longitudinal beam displacement near the central buckle of the BRB decreased with the increase in the damping constant, which proved the effectiveness of the combined application. Shi et al. [107,108] proposed a damping system with a combination of bearings, SCEDBs, and BRBs between piers and beams and found that bearings could bear the rotational displacement of the beam, the BRBs reduced the horizontal displacement, and the SCEDBs controlled the residual displacement. Based on the dynamic time-history analysis of a three-span continuous railway beam arch bridge, it was found that the combined application of SCEDBs and BRBs controlled the self-recovery ratio between 0.02 and 0.15, effectively reduced the seismic response, and controlled the residual displacement under near-fault ground motion, and that the damping rate was up to 94%. Bai et al. [109] proposed a new shock absorption system, which had a BRB and a cable restrainer on simply-supported girder bridges. The results showed that the system could control the longitudinal seismic responses of girders and transition pier. However, the specific parameters need further study.

The longitudinal arrangement of other seismic isolation devices on the superstructure can reduce the displacement of the main beam, and the transverse arrangement of BRBs on the substructure can reduce the seismic response of the piers. The bidirectional seismic isolation system can play their respective roles in the longitudinal and transverse directions and maximize the overall seismic performance of the bridge.

## 5. Conclusions and Expectations

BRBs are widely used in buildings, but only slight progress has been made in bridge engineering. Various scholars have modified the structures of traditional BRBs. Through continuous research and tests, BRBs have developed into lightweight and high-performance components. This paper provides a detailed overview of the current research status of BRBs around the world and looks to the future of BRB research:(1)The damping effect of the BRB is closely related to its yield strength, layout form, structure styles, and other parameters. Using new materials and new structures, the BRB form is simplified and easy to install and disassemble so as to meet the economic applicability of the bridge structure with replaceable components. Different parameters and arrangements of the BRB have been tested and simulated to improve the hysteresis performances of BRBs and to mitigate the responses of bridges at critical locations to select a reasonable damping solution.(2)Most of the BRB placement positions have been on pier columns, and there has been little study on whether the placement in other positions has a good damping effect on bridges. This paper proposes the idea of using BRBs in the superstructures of arch bridges and cable-stayed bridges and proves its feasibility in theory.(3)The connection nodes between BRBs and structures have an impact on structural deformation. Ensuring the stability and reliability of the nodes can fully exploit and enhance the energy dissipation of the BRB. The nodes connecting BRBs and bridges are not well studied and need to be studied independently. The node parameters need to be further determined according to the structure and BRB.(4)Compared with other damping and isolation devices, BRBs show better performances, and when used with other components simultaneously, the whole bridge will achieve a better damping effect. The location of the seismic isolation components should be arranged according to the damage control parts of the bridge, and the type should be selected according to the structure of the bridge. There are many kinds of energy dissipation devices with strong nonlinearity and different damping mechanisms, so there is a large research space for the determination of their seismic capacity.

The application of BRBs in bridge engineering requires in-depth experimental research and refined numerical simulation analysis. Based on the hysteresis performances of BRBs, their construction form should be simplified and optimized, and the seismic response characteristics of various bridge systems should be combined to make BRBs more reliable seismic isolation components. The application of SC-BRB can be considered more to control the seismic damage of key components and ensure the rapid recovery of bridges after earthquakes.

## 6. Patents

Seismic structure of long span cable-stayed bridge with buckling restrained brace (CN201620827938.5). A new transverse seismic structure system of ribbed arch bridge with buckling restrained brace (CN201320427345.6).

## Figures and Tables

**Figure 1 materials-16-02549-f001:**
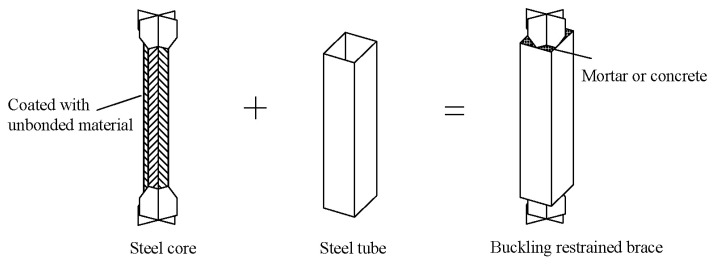
Basic composition of the buckling-restrained brace (BRB).

**Figure 2 materials-16-02549-f002:**
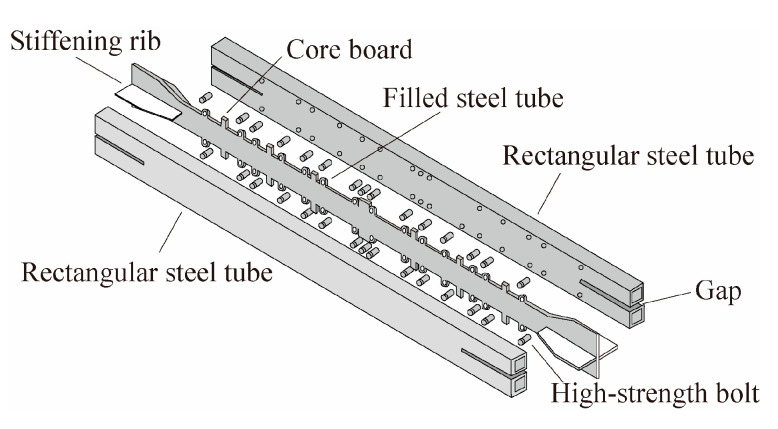
New braced structure of ductile assembled buckling-restrained brace.

**Figure 3 materials-16-02549-f003:**
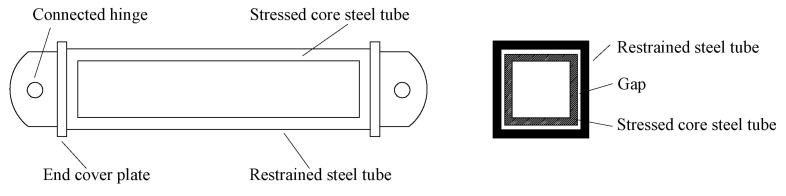
Constitution of a new type of BRB.

**Figure 4 materials-16-02549-f004:**
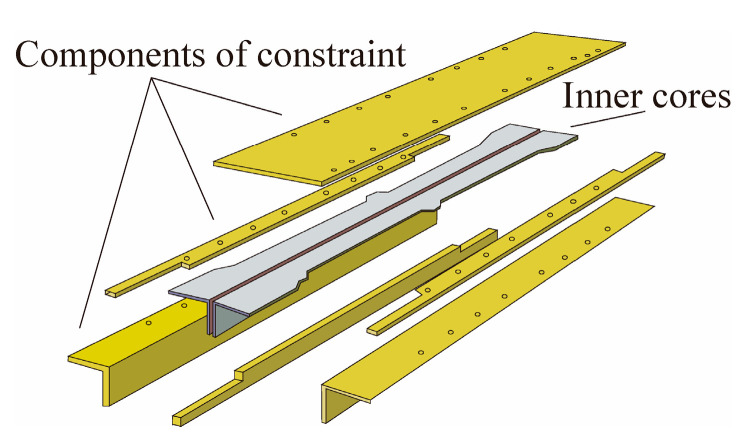
Configuration of BRB.

**Figure 5 materials-16-02549-f005:**
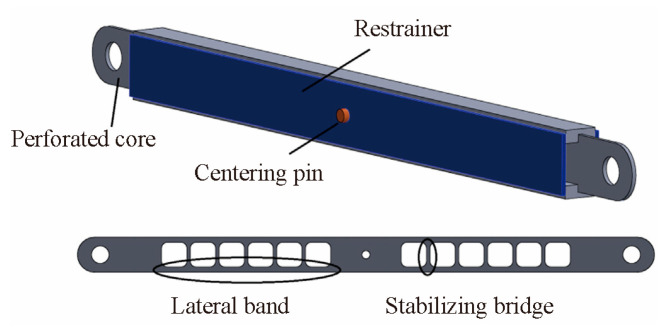
Perforated-core BRB [10]. Reproduced with permission from [X. Cahis, E], [Engineering Structures]; published by [Elsevier], [2018].

**Figure 6 materials-16-02549-f006:**
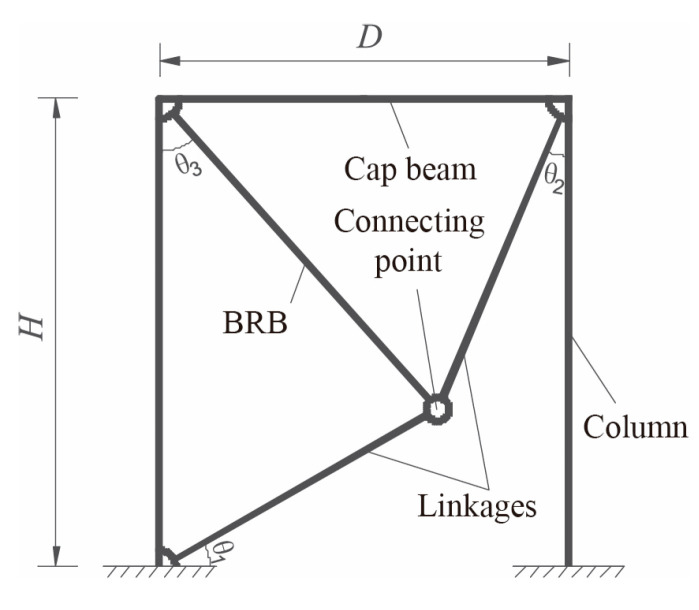
Geometric configuration of toggle BRB system [12]. Reproduced with permission from [Yan Shi], [Engineering Structures]; published by [Elsevier], [2020].

**Figure 7 materials-16-02549-f007:**
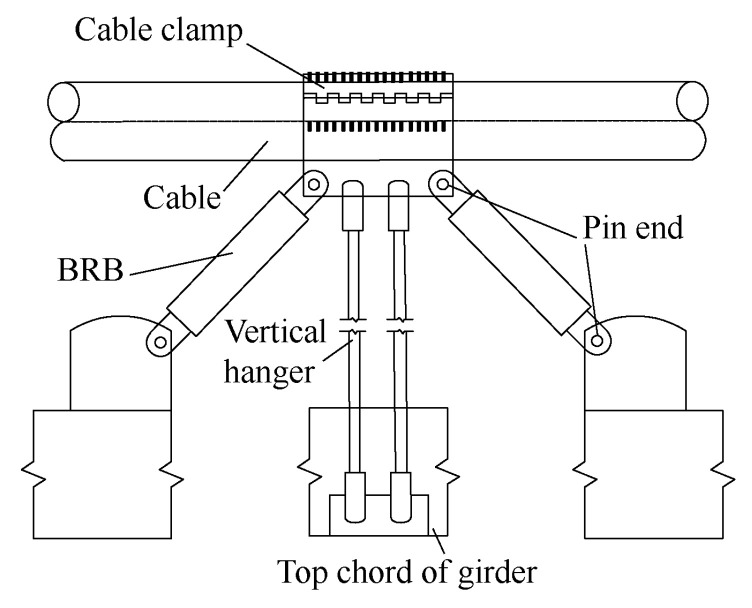
BRB central buckle.

**Figure 8 materials-16-02549-f008:**
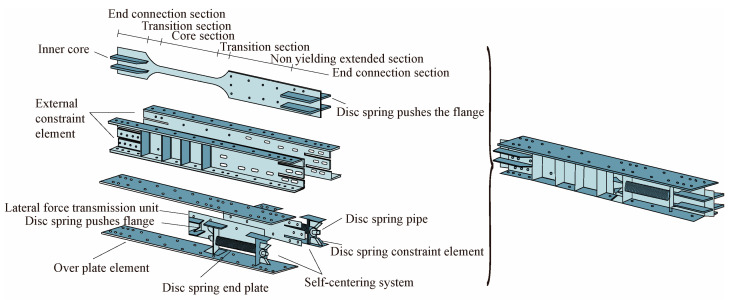
Configuration of self-centering buckling-restrained brace (SC-BRB).

**Figure 9 materials-16-02549-f009:**
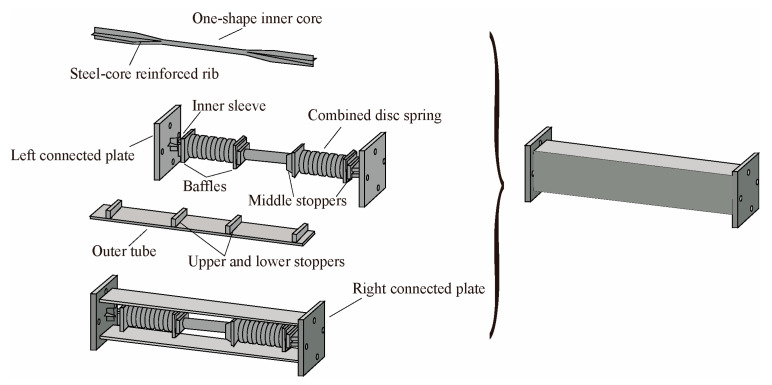
Disc-spring self-centering buckling-restrained brace (DS-SCB).

**Figure 10 materials-16-02549-f010:**
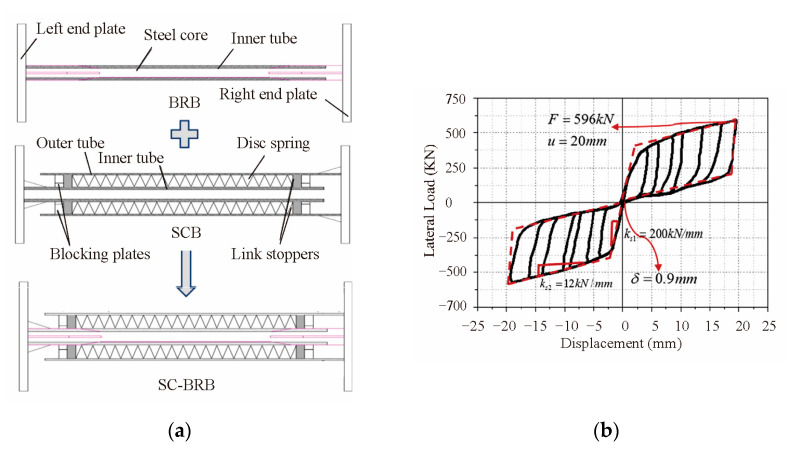
Self-centering buckling-restrained brace (SC-BRB) [30]. Reproduced with permission from [Huihui Dong], [Engineering Structures]; published by [Elsevier], [2017]. (**a**) Configuration of SC-BRB system; (**b**) Hysteretic curves of SC-BRB.

**Figure 11 materials-16-02549-f011:**
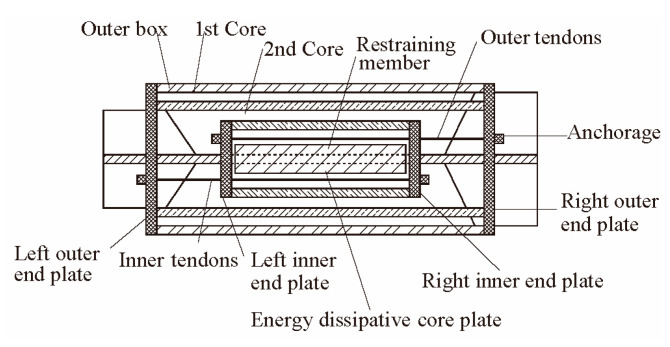
Proposed dual-core SC-SBRB [31]. Reproduced with permission from [Chung-Che Chou], [Engineering Structures]; published by [Elsevier], [2016].

**Figure 12 materials-16-02549-f012:**
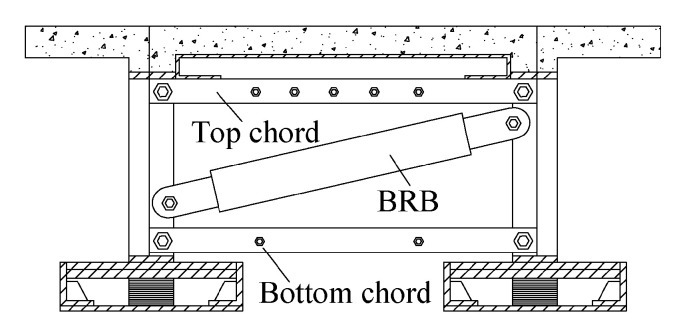
End cross frames in bridge model using BRBs.

**Figure 13 materials-16-02549-f013:**
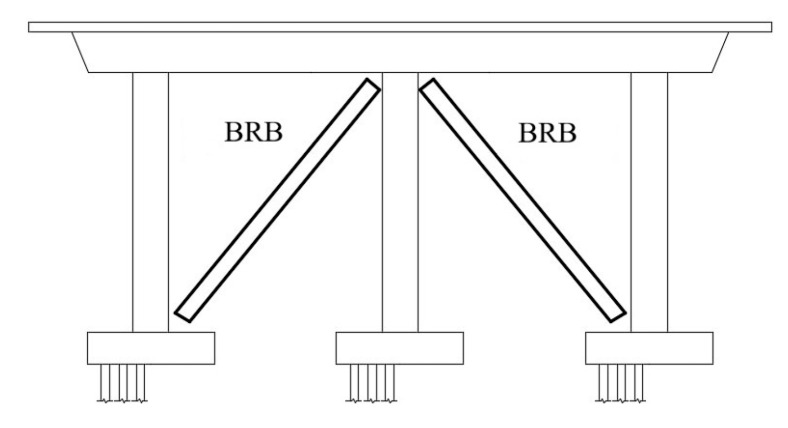
The layout of BRBs between piers and cover beam.

**Figure 14 materials-16-02549-f014:**
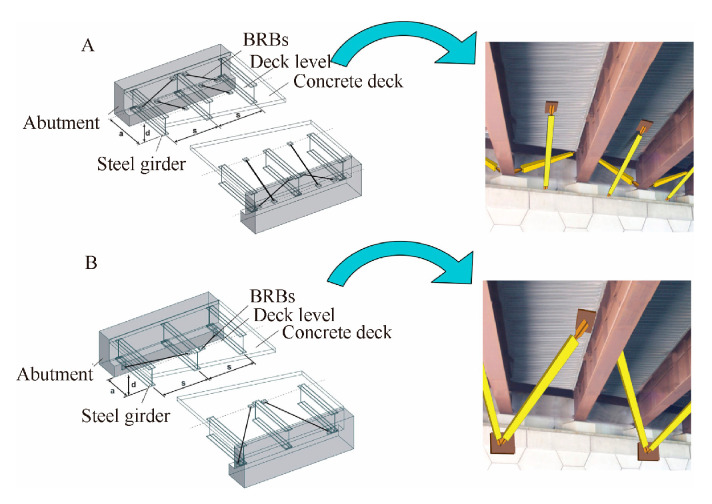
Two schemes of bridge end diaphragm reconstruction [62]. Reproduced with permission from [Oguz C], [Engineering Structures]; published by [Elsevier], [2009]. (**A**) Scheme A; (**B**) Scheme B.

**Figure 15 materials-16-02549-f015:**
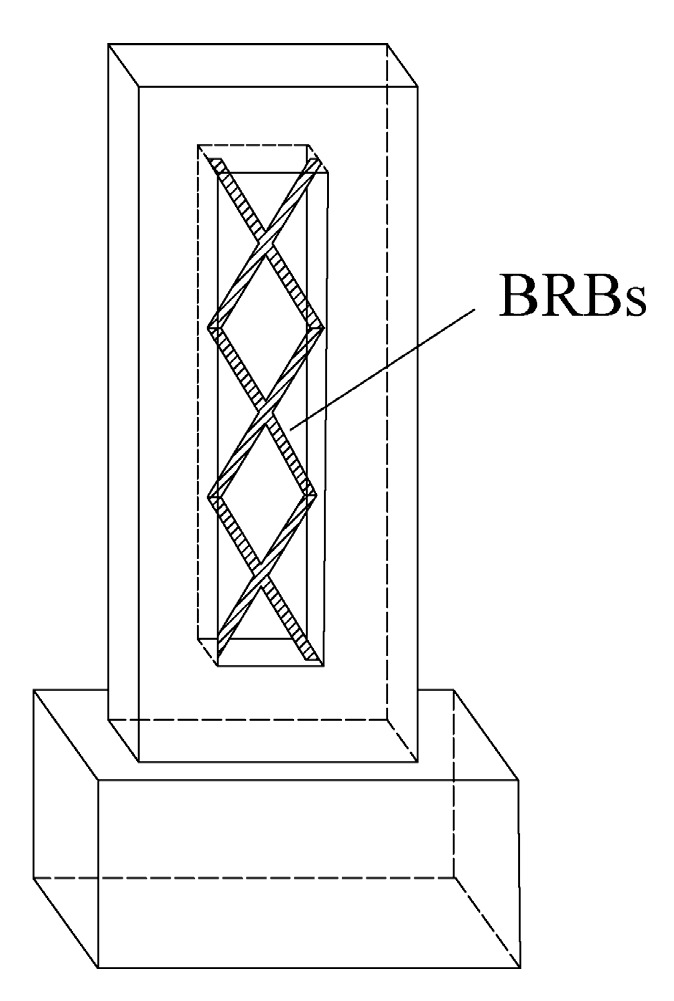
The layout of BRBs across the section of pier.

**Figure 16 materials-16-02549-f016:**
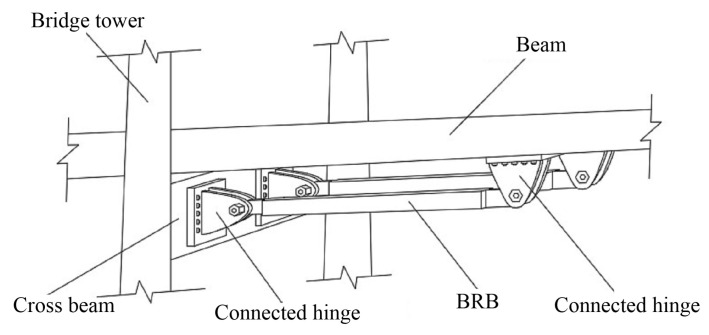
Schematic of BRBs in cable-stayed bridge.

**Figure 17 materials-16-02549-f017:**
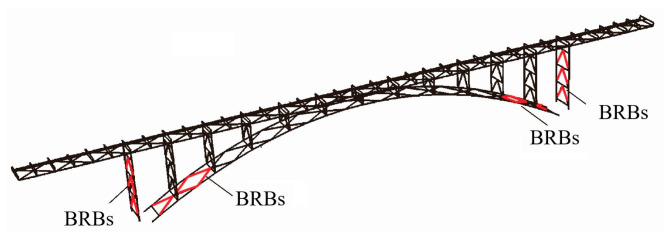
Schematic of BRBs in arch bridge [70]. Reproduced with permission from [Hanbin Ge], [Earthquake Engineering and Structural Dynamics]; published by [John Wiley and Sons], [2005].

**Figure 18 materials-16-02549-f018:**
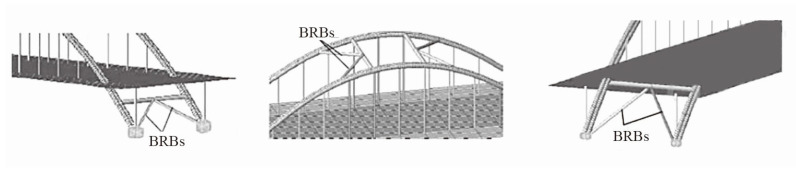
Schematic of BRBs in arch bridge.

**Figure 19 materials-16-02549-f019:**
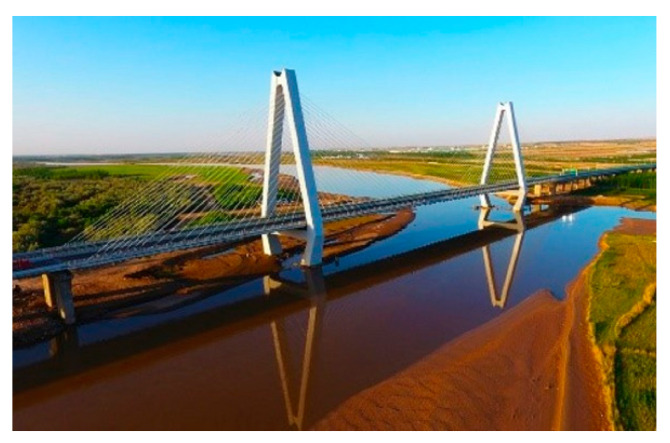
Yong Ning Yellow River Bridge [79].

**Figure 20 materials-16-02549-f020:**
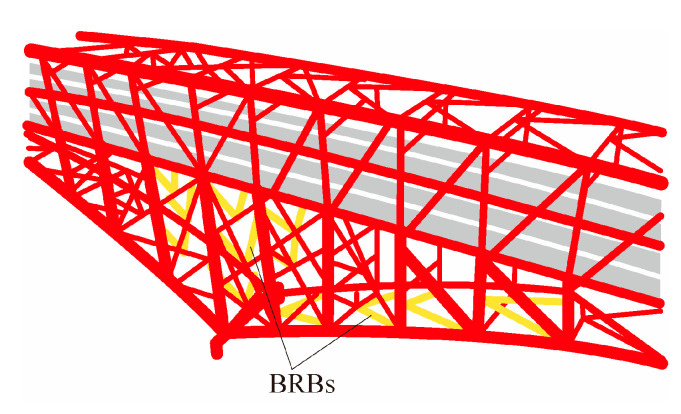
Schematic of BRBs in Monito Bridge.

**Figure 21 materials-16-02549-f021:**
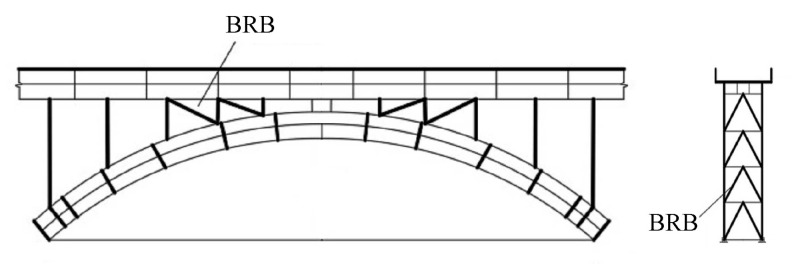
BRB layout of Wangdu Bridge.

**Figure 22 materials-16-02549-f022:**
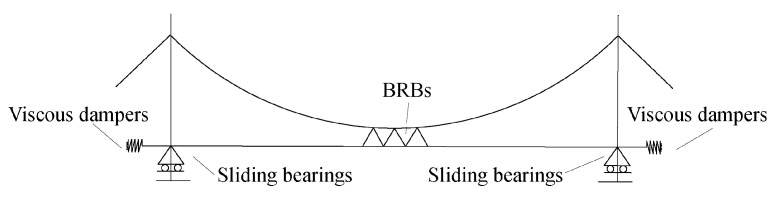
BRBs as the central buckle in Luding Dadu River Bridge.

**Figure 23 materials-16-02549-f023:**
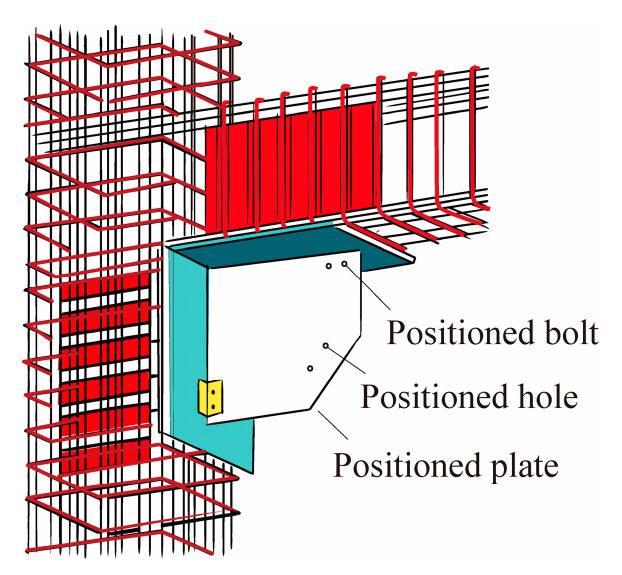
Details of sliding gusset connection.

**Figure 24 materials-16-02549-f024:**
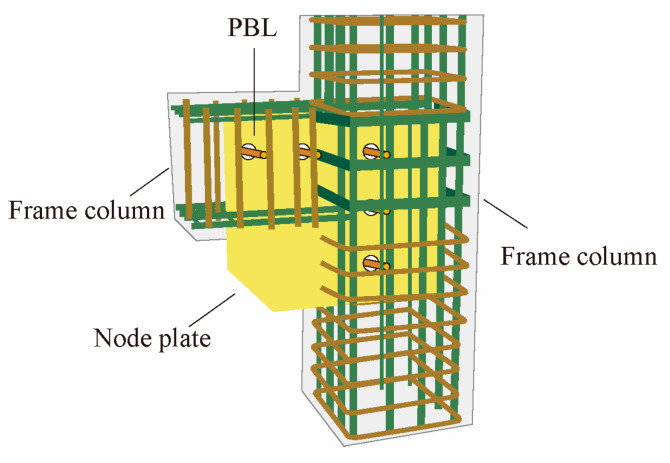
Schematic diagram of PBL gusset connection.

**Figure 25 materials-16-02549-f025:**
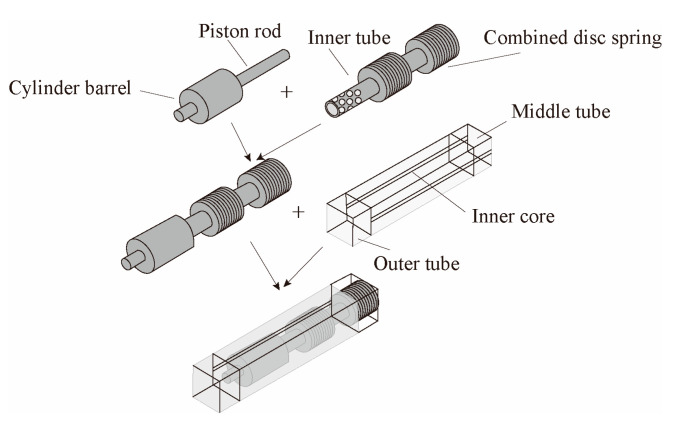
Composition of lock-up self-centering buckling-restrained brace (LU-SC-BRB).

**Table 1 materials-16-02549-t001:** Summary of advantages and disadvantages of different BRBs.

Types		Main Advantages					Main Disadvantages
		High Energy Dissipating Capacity	Light Weight	Simple Construction	High Load-Carrying Capacity and Stiffness	Little Residual Displacement	Friction
Materials	SMA inner core	✓	✓			✓	
	All-steel (TTC-BRB, SBED)	✓	✓		✓		
	MFSC-BRB		✓		✓		
	Aluminum alloy inner core		✓				
	Lightweight aggregate concrete filling materials		✓	✓			✓
Configurations	DA-BRB	✓			✓		
	DRSSP-BRB	✓		✓			
	TBRB	✓		✓			
	PCBRB	✓	✓	✓	✓		
	A toggle BRB system	✓					
	DY-BRB	✓			✓		
	BRB central buckle	✓			✓		
	SC-steel-BRB	✓					
	DS-SCB	✓				✓	✓
	SC-BRB					✓	
	SC-sandwiched-BRB	✓				✓	

**Table 2 materials-16-02549-t002:** Summary of BRBs-installed on the bridges.

Bridges	Location (BRB)	Advantages	Disadvantages
Curved Beam Bridge (girder bridge)	Between the cover beam and pier	BRB improved the seismic performance of bridges [53,54].	Residual deformation
A reinforced-concrete anti-bending bridge (girder bridge)	Between the piers	BRB improved the displacement ductility of the structure and controlled the damage of the existing vulnerable reinforced concrete bent. [55,56]	-
A curved high-pier bridge (girder bridge)	Between tie beams	BRB could effectively reduce the seismic vulnerability and improve the performance of the bridge [57].	BRBs only mitigated the local seismic demands.
A box-girder bridge (girder bridge)	Between the cover beam and the pier	BRB could reduce the bending moments, displacements, and the potential damage [58].	-
A post-tensioned bridge (girder bridge)	Between the cap beam and the footings	BRB could make bridge restore quickly [59,60].	-
A double-column girder bridge (girder bridge)	Between the piers	BRB could improve the transverse stiffness of the bridge and energy dissipation capacity [61].	Residual deformation
A straight steel bridge (girder bridge)	On the end diaphragm	BRB could effectively resist the longitudinal and transverse seismic forces [62].	-
A single-span steel slab-on-girder bridge (girder bridge)	On the end diaphragm	BRB had the ability to withstand bidirectional displacement demands [63,64].	Temperature changes had an effect on BRBs.
Long-span cable-stayed bridges	Between the piers	BRB improved the energy dissipation capacity of the auxiliary pier [66].	The higher the yield strength of BRB, the less early energy consumption.
A concrete cable-stayed bridge	Across the section of pier	BRB improved the transverse seismic performance of the side span pier column [67].	BRB form of section would affect.
A cable-stayed bridge	Between the cross beam and the beam	BRB greatly reduced the bending moment of the tower (pier) [68,69].	The relative displacement control of the support and the top of the tower by BRB is limited.
A steel arch bridge	On the columns and main arch	BRBs improved the seismic performance of the bridge [70,71].	Residual deformation
A long-span steel truss railway arch bridge	At the bottom and top of the chord planes	BRB reduced the internal forces at the bottom section of the columns on each arch [72].	The seismic response of some bars in the arch would be increased.
Arch bridges	Replace beams or transverse braces	BRB improved the energy dissipation capacity and damping effect [73,74].	-
A steel truss arch bridge	Replace a portion of the normal bars	BRB could reduce the internal forces and displacements of the arch ribs [75].	-
Light flexible arch bridges	Between bent piers	BRB could improve the lateral seismic performance of bridges [76].	The damage of high piers might increase.

## Data Availability

Not applicable.

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
