# Peer review of "Research on the Application of BRBs in Seismic Resistance of Bridge"

_materials, 2023, doi:10.3390/ma16072549_

Round 1

Reviewer 1 Report (New Reviewer)

The paper entitled “Research on the application of BRBs in seismic resistance of bridge” provides a very interesting research work about the behavior and the effectiveness of the protection provided to a bridge structural systems against earthquake excitations, by implementing BRB elements. The manuscript is well written, easy to read and provides very useful information. However, a number of revisions are suggested, in order to increase the value of the manuscript.

1.      Page 12 – Line 356: “Ordinary seismic isolation bearings are not suitable for special bridge types such as long-span arch bridges, so BRBs are novel damping structures for arch bridges.”

Page 17 – Line 481: “BRBs as braced and energy-dissipating dampers have demonstrated superior 481 seismic performances compared to other seismic isolation devices.”

The article does not provide general information about ordinary seismic isolation devices, and most importantly does not compare explicitly isolators to BRB from a theoretical point of view. Thus, Authors are suggested to deepen the Introduction section, by providing additional descriptions of the ordinary seismic isolation devices: this would increase the quality of the manuscript and would help the reader in understanding the effective advantages in using the proposed BRB strategy. Among the other, the following research works are suggested:

http://dx.doi.org/10.1016/j.engstruct.2017.04.032

https://doi.org/10.3390/vibration5020017

2.      Page 5 – Line 141: “Christopulos et al. [18] first used the internal viscous resistance of the component to consume energy and proposed the concept of a self-centering energy dissipation brace (SCEDB) that was composed of composite materials. The SCEDB for steel structures was verified to have little residual deformation and did not usually change 144 after an earthquake [19,20].”. Prof. Constantin Christopoulos is a pioneer in the development of self-centering braces. Authors are encouraged to provide additional details on the topic, also about rocking systems coupled with dissipative braces.

3.      In the whole review document a number of technologies are shown and discussed. However, no indication about possible design procedure are provided. Are there for any or for all the commented devices some rules to design geometrical and mechanical characteristics of the devices? These additional details would increase the overall quality of the manuscript.

Author Response

Dear reviewer:

Thank you for the advice. We are very grateful to you for your suggestions on the revision of this article. In this round of revision, we focused our efforts strongly on the points made in your letter. And we took seriously and addressed in some way each of your comments. Your comments have again stimulated changes we feel further improved the paper.

First of all, the title of this paper is “Research on the application of BRBs in seismic resistance of bridge”. The corresponding changes to the resubmitted manuscript are highlighted in yellow.

Point 1: The article does not provide general information about ordinary seismic isolation devices, and most importantly does not compare explicitly isolators to BRB from a theoretical point of view. Thus, Authors are suggested to deepen the Introduction section, by providing additional descriptions of the ordinary seismic isolation devices: this would increase the quality of the manuscript and would help the reader in understanding the effective advantages in using the proposed BRB strategy. Among the other, the following research works are suggested:

http://dx.doi.org/10.1016/j.engstruct.2017.04.032

https://doi.org/10.3390/vibration5020017

Response 1: We try to retrieve the relevant typical references reflecting the research of BRB and isolators, a total of 5 articles [90-95] in Section 4.1. We introduce the theoretical difference between isolators and BRB and we put the focus on BRB more.

Point 2: Page 5 – Line 141: “Christopulos et al. [18] first used the internal viscous resistance of the component to consume energy and proposed the concept of a self-centering energy dissipation brace (SCEDB) that was composed of composite materials. The SCEDB for steel structures was verified to have little residual deformation and did not usually change 144 after an earthquake [19,20].”. Prof. Constantin Christopoulos is a pioneer in the development of self-centering braces. Authors are encouraged to provide additional details on the topic, also about rocking systems coupled with dissipative braces.

Response 2: We have added additional details on the topic of SCEDB in Section 2.3. We described the structures and working principle of SCEDB. And we added rocking systems with dissipative braces [24-27] in Section 2.3.

Point 3: In the whole review document a number of technologies are shown and discussed. However, no indication about possible design procedure are provided. Are there for any or for all the commented devices some rules to design geometrical and mechanical characteristics of the devices? These additional details would increase the overall quality of the manuscript.

Response 3: In bridge engineering, there is no common design procedure for all types bridges. However, there are some rules to design geometrical and mechanical characteristics of the devices, the rules are supplied in the end of Chapter 2.

Reviewer 2 Report (Previous Reviewer 2)

Please:

- improve / revise further English and typos;

- improve quality of figures;

- clarify in the figure captions the literature source for each one of them

Author Response

Dear reviewer:

Thank you for the advice. We are very grateful to you for your suggestions on the revision of this article. In this round of revision, we focused our efforts strongly on the points made in your letter. And we took seriously and addressed in some way each of your comments. Your comments have again stimulated changes we feel further improved the paper.

First of all, the title of this paper is “Research on the application of BRBs in seismic resistance of bridge”. The corresponding changes to the resubmitted manuscript are highlighted in yellow.

Point 1: improve / revise further English and typos;

Response 1: We have read the whole article carefully many times and used the professional English software to check the article. We think that English and typos have been improved.

Point 2: improve quality of figures;

Response 2: We have used professional drawing software (Adobe Illustrator) to revise all figures and downloaded the origin figures to revise. Figure 1 was drawn again by CAD. The words of Figure 6 were edited. The watermark from the Figure 18 was removed. Figure 20 was from the references [78] in 2005 and not clear, we changed the words. The outermost border of Figure 26 has been deleted. We have tried our best to improve quality of figures and each figure was in high resolution.

Point 3: clarify in the figure captions the literature source for each one of them.

Response 3: We have added the corresponding document number at the end of each figure title. The figure 1 and 21 was drawn by CAD. Figure 19 was downloaded from the internet and the link of the figure was in the supplementary materials part.

Reviewer 3 Report (New Reviewer)

This paper presents a review on the application of BRB braces in bridges that can be useful for the researchers in this field. I suggest to accept and publish this paper after addressing the following recommendations:

1- Regarding the all-steel BRBs, please add the following papers to Section 2.2:

a. Akbari Hamed A, Hashemi SS. Parametric study on the structural performance of ordinary, bamboo-shaped and triple-truss confined all-steel BRBs with a circular core cross-section. Asian Journal of Civil Engineering. 2023 Jan 17:1-25.

b. Guo YL, Zhou P, Wang MZ, Pi YL, Bradford MA, Tong JZ. Experimental and numerical studies of hysteretic response of triple-truss-confined buckling-restrained braces. Engineering Structures. 2017 Oct 1;148:157-74.

c. Wang CL, Liu Y, Zhou L. Experimental and numerical studies on hysteretic behavior of all-steel bamboo-shaped energy dissipaters. Engineering Structures. 2018 Jun 15;165:38-49.

2- Please fit Figures 9-10 and Tables 1 and 2 to the margins of the page. It is suggested to rotate the mentioned tables to present them in a landscape format.

3- In Section 3, it is recommended to add a brief explanations along with the corresponding figures about different types of the mentioned bridges.

4- As the importance of Section4, it is strongly recommended to add more references about "Comparison of BRB with other seismic isolation components" and "Combining BRBs with other seismic isolation components". 

Author Response

Dear reviewer:

Thank you for the advice. We are very grateful to you for your suggestions on the revision of this article. In this round of revision, we focused our efforts strongly on the points made in your letter. And we took seriously and addressed in some way each of your comments. Your comments have again stimulated changes we feel further improved the paper.

First of all, the title of this paper is “Research on the application of BRBs in seismic resistance of bridge”. The corresponding changes to the resubmitted manuscript are highlighted in yellow.

Point 1: Regarding the all-steel BRBs, please add the following papers to Section 2.2…

Response 1: We have added all-steel BRBs papers [16-18] and made introductions of them in Section 2.2. We concluded them into all-steel BRB in Table 1, and all steel BRBs showed excellent performance in load carrying capacity and energy dissipating capacity.

Point 2: Please fit Figures 9-10 and Tables 1 and 2 to the margins of the page. It is suggested to rotate the mentioned tables to present them in a landscape format.

Response 2: We have fit Figure 9-10 to the margins and adjusted the Tables 1-2 to meet the requirements of the literature format.

Point 3: In Section 3, it is recommended to add a brief explanation along with the corresponding figures about different types of the mentioned bridges.

Response 3: We have added the corresponding bridge type in Table 2, and some types of bridges have been described in the first column of the table.

Point 4: As the importance of Section4, it is strongly recommended to add more references about "Comparison of BRB with other seismic isolation components" and "Combining BRBs with other seismic isolation components".

Response 4: We try to retrieve the relevant typical references reflecting the research of BRB and isolators, a total of 5 articles [90-95] in Section 4.1.

Round 2

Reviewer 1 Report (New Reviewer)

Authors have significantly increased the quality of the manuscript and all the comments have been addressed. Therefore, the article can be accepted for publication in its revised form.

Author Response

Dear Reviewer:

Thank you for the advice. We are very grateful to you for your suggestions on the revision of this article. First of all, the title of this paper is “Research on the application of BRBs in seismic resistance of bridge”. We have read the whole article carefully many times and used the professional English software to check the article. We think that English and typos have been improved. On the advice of academic editors, we also added the new contents and the corresponding changes to the resubmitted manuscript are highlighted in yellow.

This manuscript is a resubmission of an earlier submission. The following is a list of the peer review reports and author responses from that submission.

Round 1

Reviewer 1 Report

This is a good attempt to gather and present in a review form the application of BRB in bridge engineering. Before final acceptance of the paper the authors should perform once again a thorough reference check in SCOPUS as some papers published in 2022 do not exist in the list of references. Please also look into the proceedings of the World Conference on Earthquake Engineering (after the year 2000) that are available online.

A thorough language should be also performed.

Author Response

Dear reviewer:

Thank you for the positive feedback and conditionally accepting our paper. We are very grateful to you for your suggestions on the revision of this article. In this round of revision, we focused our efforts strongly on the points made in your letter. Although we replied directly to you and focused our explanations on points raised in your letter, we took seriously and addressed in some way each of your comments. Your comments have again stimulated changes we feel further improved the paper.

First of all, the title of this paper is “Research on the application of BRBs in seismic bridge structures”. Our responses are given directly in a different color (Red).

Point 1: This is a good attempt to gather and present in a review form the application of BRB in bridge engineering. Before final acceptance of the paper the authors should perform once again a thorough reference check in SCOPUS as some papers published in 2022 do not exist in the list of references. Please also look into the proceedings of the World Conference on Earthquake Engineering (after the year 2000) that are available online.

 Response 1: According to the opinions, we added more references about BRBs in 2022 and 2023. When similar bridge types appear, only the earliest test or simulation was used as the representative type. We chose the main topic of BRBs types before, and we added the new material of BRBs [13,23,25,27,50,63].

Point 2: A thorough language should be also performed.

Response 2: We tried our best to improve the manuscript and made some changes to the manuscript. We invited professional institution of English language to help polish our article.

We would like also to thank you for allowing us to resubmit a revised copy of the manuscript. The revised parts of the manuscript are highlighted in blue.

We appreciate for your warm work earnestly. Once again, thank you very much for your comments and suggestions.

Sincerely,

Xiaoli Li, Jina Zou, Dongsheng Wang, Yuemin Zhao

Reviewer 2 Report

This manuscript is presented as "review" on the use of BRBs for seismic resistant bridge structures.

The topic is relevant but the document does not reflect the current state of art on the topic and needs major extension in literature and also on discussion of collected documents.

Author Response

Dear reviewer:

Thank you for the positive feedback and conditionally accepting our paper. We are very grateful to you for your suggestions on the revision of this article. In this round of revision, we focused our efforts strongly on the points made in your letter. Although we replied directly to you and focused our explanations on points raised in your letter, we took seriously and addressed in some way each of your comments. Your comments have again stimulated changes we feel further improved the paper.

First of all, the title of this paper is “Research on the application of BRBs in seismic bridge structures”. Our responses are given directly in a different color (Red).

Point 1: The topic is relevant but the document does not reflect the current state of art on the topic and needs major extension in literature and also on discussion of collected documents.

 Response 1: According to the opinions, we added more references about BRBs in 2022 and 2023. We added a new chapter 2.4 titled “Study on new materials of buckling-restrained brace” [23-26]. There are not too many projects and references about BRB in bridge engineering, we only quoted some typical references to prove the high energy dissipating capacity of BRB and necessary application in bridges, such as reference [1,14,18,20,35,39,51,56,69]. And we also added new references [13,27,50,63].

We would also like to thank you for allowing us to resubmit a revised copy of the manuscript. The revised parts of the manuscript are highlighted in blue.

We appreciate for your warm work earnestly. Once again, thank you very much for your comments and suggestions.

Sincerely,

Xiaoli Li, Jina Zou, Dongsheng Wang, Yuemin Zhao

Reviewer 3 Report

The objective of the present work is to elaborate on the BRB’s application as a seismic equipment device in bridge structures. The classical and new generations of BRB’s introduced. The reviewed manuscript was well-written and organized. I highly recommend to the authors increase more references to improve the literature review. i.e. application of smart materials in BRB’s:

* Application and modeling of Shape-Memory Alloys for structural vibration control: State-of-the-art review, 2022

* Probabilistic Assessment the Seismic Collapse Capacity of Buckling-Restrained Braced Frames Equipped with Shape Memory Alloys, 2021.

or probabilistic safety studies about bridge BRB's and etc. 

Author Response

Dear reviewer:

Thank you for the positive feedback and conditionally accepting our paper. We are very grateful to you for your suggestions on the revision of this article. In this round of revision, we focused our efforts strongly on the points made in your letter. Although we replied directly to you and focused our explanations on points raised in your letter, we took seriously and addressed in some way each of your comments. Your comments have again stimulated changes we feel further improved the paper.

First of all, the title of this paper is “Research on the application of BRBs in seismic bridge structures”. Our responses are given directly in a different color (Red).

Point 1: The objective of the present work is to elaborate on the BRB’s application as a seismic equipment device in bridge structures. The classical and new generations of BRB’s introduced. The reviewed manuscript was well-written and organized. I highly recommend to the authors increase more references to improve the literature review. i.e. application of smart materials in BRB’s:

* Application and modeling of Shape-Memory Alloys for structural vibration control: State-of-the-art review, 2022 https://www.eng.uwo.ca/civil/faculty/youssef_m/docs/SMA%20Bridge.pdf

* Probabilistic Assessment the Seismic Collapse Capacity of Buckling-Restrained Braced Frames Equipped with Shape Memory Alloys, 2021.

or probabilistic safety studies about bridge BRB's and etc.

Response 1: According to the opinions, we add more references about SMA BRBs in the revised manuscript. We added a new chapter 2.4 titled “Study on new materials of buckling-restrained brace” [23-26]. We also added a new material BRB, which is a ribbed glass fiber reinforced polymer (GFRP) rectangular tube used as the restraint unit [27]. They are suitable for high-rise buildings and bridge engineering. Probabilistic safety studies about bridge BRB were studied in the paper [41-43].

We would also like to thank you for allowing us to resubmit a revised copy of the manuscript. The revised parts of the manuscript are highlighted in blue.

We appreciate for your warm work earnestly. Once again, thank you very much for your comments and suggestions.

Sincerely,

Xiaoli Li, Jina Zou, Dongsheng Wang, Yuemin Zhao

Reviewer 4 Report

The paper is a review of the application of buckling-restrained braces (BRBs) in bridge engineering.

Number of references is sufficient. However, there is a lack of presented knowledge using tables and graphs, comparison of advantages and disadvantages, effectiveness etc.

In introduction should be discussed advantages and disadvantages of the BRBs in comparison with alternative options.

Chapter 2 Different types of BRBs should be systematically present in a table with layouts,  references, advantages and disadvantages. Instead of long text with a short summary of individual articals, authors should discuss similarities, differences, advantages, disadvantages...

Chapter 3 Systematization  is required: outline positions of bbr, advantages and disadvantages, sketches + comparative comparison with advantages and disadvantages.

Chapter 3.5 BRB layouts of all existing bridges presented in the paper should be added as figures.  

Author Response

Dear reviewer:

Thank you for the positive feedback and conditionally accepting our paper. We are very grateful to you for your suggestions on the revision of this article. In this round of revision, we focused our efforts strongly on the points made in your letter. Although we replied directly to you and focused our explanations on points raised in your letter, we took seriously and addressed in some way each of your comments. Your comments have again stimulated changes we feel further improved the paper.

First of all, the title of this paper is “Research on the application of BRBs in seismic bridge structures”. Our responses are given directly in a different color (Red).

Point 1: Number of references is sufficient. However, there is a lack of presented knowledge using tables and graphs, comparison of advantages and disadvantages, effectiveness etc.

Response 1: We sincerely accepted this problem that we did not have more tables and graphs, comparison of advantages and disadvantages, effectiveness etc. Therefore, we added new graphs about the typical layout of BRB in chapter 3 (Figure 12,14,16,17,19,23), table1 in chapter 2, and table2 in chapter 3.

Point 2: In introduction should be discussed advantages and disadvantages of the BRBs in comparison with alternative options.

Response 2: We briefly added advantages and disadvantages of the BRBs in comparison with alternative options in introduction. And the comparison and explanation were introduced in detail of chapter 4.

Point 3: Chapter 2 Different types of BRBs should be systematically present in a table with layouts, references, advantages and disadvantages. Instead of long text with a short summary of individual articles, authors should discuss similarities, differences, advantages, disadvantages.

Response 3: The layouts, advantages and disadvantages were listed in Table 1 of chapter 2. In summary, the construction of traditional BRBs has been continuously updated and developed. New BRBs have high energy dissipating capacity and other advantages, as shown in Table 1. BRBs also have other disadvantages. Only the most influential friction is listed in the Table 1.

Point 4: Chapter 3 Systematization is required: outline positions of brb, advantages and disadvantages, sketches + comparative comparison with advantages and disadvantages.

Response 4: We added new graphs about the layout of BRB (Figure 12,14,16,17). We added advantages and disadvantages of BRB in the bridge in table 2 of chapter 3.

Point 5: Chapter 3.5 BRB layouts of all existing bridges presented in the paper should be added as figures.

Response 5: We have completed the BRB layouts of Monito Bridge in Figure 19 of chapter 3.5. Yong Ning Yellow River Bridge did not have the BRB layouts figures, but we described detailed BRB layouts in chapter 3.5. We also tried our best to improve the manuscript and made some changes to the manuscript.

We would also like to thank you for allowing us to resubmit a revised copy of the manuscript. The revised parts of the manuscript are highlighted in blue.

We appreciate for your warm work earnestly. Once again, thank you very much for your comments and suggestions.

Sincerely,

Xiaoli Li, Jina Zou, Dongsheng Wang, Yuemin Zhao

Round 2

Reviewer 1 Report

The paper is accepted for publication and can be moved to production.

Reviewer 2 Report

The original document has been revised but still lacks of organized and consistent discussion of existing works, as it would be expected from a review article.

Quality of figures is also very poor.

References do not reflect the state of art.

Reviewer 4 Report

The paper is improved and can be accepted in present form.